# Heat Pump Drying of Kelp (*Laminaria japonica*): Drying Kinetics and Thermodynamic Properties

Qian Zhang [1,2,3], Shiyu Li [2], Minqi Zhang [2], Gang Mu [2,3], Xiuchen Li [2,3], Guochen Zhang [2,3,*] and Shanbai Xiong [1,*]

1   College of Food Science and Technology, Huazhong Agricultural University, Wuhan 430070, China; zhangqian@dlou.edu.cn
2   Technology Innovation Center of Marine Fishery Equipment in Liaoning, Dalian Ocean University, Dalian 116023, China; lishiyu1016@163.com (S.L.); zhangminqi19960124@163.com (M.Z.); mugang@dlou.edu.cn (G.M.); lxc@dlou.edu.cn (X.L.)
3   Key Laboratory of Environment Controlled Aquaculture Ministry of Education, Dalian Ocean University, Dalian 116023, China
*   Correspondence: zhangguochen@dlou.edu.cn (G.Z.); xiongsb@mail.hzau.edu.cn (S.X.); Tel.: +86-136-1092-3768 (G.Z.); +86-139-7103-9012 (S.X.)

**Abstract:** The main objective of this research is the study of the drying kinetics and thermodynamic properties of kelp using heat pump drying technology. The effects of the independent variables of temperature (20–50 °C), air velocity (0.3–1.3 m/s), humidity (20–50%), and thickness (0.8–4.2 mm) on the drying time, moisture uniformity, effective moisture diffusivity ($D_{eff}$), activation energy ($E_a$), enthalpy ($\Delta H$), entropy ($\Delta S$), and Gibbs free energy ($\Delta G$) were investigated. The results show that the Page model was effective in describing the moisture content change of kelp during heat pump drying. The $D_{eff}$ varied from $1.00 \times 10^{-11}$ to $13.00 \times 10^{-11}$ m$^2$/s and the temperature, air velocity, humidity, and thickness had significant effects on drying time and moisture uniformity. Higher temperature and air velocity with proper humidity shortened the drying time and lessened the influence of thickness on moisture uniformity. The $E_a$ (16.38–26.66 kJ/mol) and $\Delta H$ (13.69–24.22 kJ/mol) were significantly increased by thickness. When the temperature was 40 °C, air velocity 1.3 m/s, and air humidity 40%, the moisture content was reduced to 18% in 5 h, with a homogeneous moisture content. This study clarifies the regularity of moisture change inside kelp and provides a theoretical reference for the development of macroalgae drying technology.

**Keywords:** heat pump drying; kelp; kinetics; thermodynamic properties; moisture uniformity





## 1. Introduction

Trade in aquatic plants increased from USD 65 million in 1976 to more than USD 1.3 billion in 2018 [1]. Kelp (Laminaria japonica) accounts for more than 50% of macroalgae production, and is rich in iodine, vitamins, minerals, proteins, fatty acids, and other physiologically active ingredients [2,3]. These nutrients provide kelp with high edible value and special efficacy in lowering blood pressure, as well as possessing anti-tumor, anti-radiation, and anti-virus properties. They can also improve immune function and so on [4]. However, kelp's moisture content is generally above 90%, which shortens its shelf-life to only 2–5 days [5]. Currently, drying is an important method to extend kelp's shelf-life and provide added value. The major dehydration method of kelp is nature drying, which is time-consuming and negatively affects the quality. Foscarini et al. [6] found that drying Eucheuma seaweed took 3–5 days with approximately 8–9 h of sunshine per day. Ling et al. [7] also reported that drying Kappaphycus alvarezii in sunlight took 3–4 days and significantly reduced the total phenolic, flavonoid, anthocyanin, and carotenoid content compared with seaweeds dried in ovens.

Heat pump drying offers obvious advantages in terms of drying efficiency and product quality [8]. The heat pump system consists of a compressor, evaporator, condenser, and expansion valve, etc. The refrigerant, such as R134a, R410, R717, or R744 [9], is evaporated into gas in the evaporator, driven by a compressor, which absorbs a large amount of heat energy from the air. The gaseous refrigerant is compressed into a high-temperature, high-pressure gas, which then enters the condenser to release heat to the drying medium (air), repeating the cycle until the material is dried. Heat pump drying is suitable for the large-scale processing of aquatic products [10] and has been successfully applied to kelp knot [11], scallop adductors [12], squid fillets [13], banana slices [14], grape pomace [15], and lignite [16].

In recent years, studies on kelp drying have mainly focused on its physical characteristics and nutrients [17,18], while only a few studies have investigated the heat pump drying kinetics and thermodynamic properties, which explain the inner mechanism of heat and mass transfer and offer necessary theoretical parameters for drying technology and product quality improvement [19]. Kelp plants are generally 2–5 m in length and 20–50 cm in width [20]; the huge leaves are uneven in thickness and the moisture content of different parts is inconsistent. Therefore, drying non-uniformity and quality degradation under improper drying processes are likely, but the uniformity of the moisture content and the effect of thickness on the drying characteristics have rarely been reported. Hu et al. [11] studied the heat pump drying kinetics characteristics of kelp knot and found that the Page model could effectively fit the changing rule of moisture content with time, but the drying of spreading kelp leaves and their kinetics and thermodynamic properties were unclear. Tunckal et al. [14] fitted the moisture ratio of banana slices during heat pump drying with the Midilli and Kucuk model, and the effective moisture diffusivity values were calculated to be between $1.12 \times 10^{-10}$ and $1.64 \times 10^{-10}$ m$^2$/s over a temperature range of 37–42 °C. Almeida et al. [21] calculated the effective moisture diffusivity, activation energy, and the thermodynamic properties of Achachairu (*Garcinia humilis*) peels under the drying process. Their results provided important information about the moisture migration and the required energy during the process. A heat pump-assisted drying process can reduce the energy demand by 84% compared to traditional drying using fossil fuels, but possibly results in up to a 69% longer drying time due to the higher humidity [9]. To produce consistent-quality seaweed and shorten the drying time, Sarbatly et al. [8] studied the heat pump drying kinetic and thermodynamic properties of *Eucheuma spinosum*, and found that the difference in algae enthalpy values used in each experiment was up to 1.74 times, but the reason was not explained. Silva et al. [22] recommended standardizing and specifying the leaf thickness measuring points in the thermodynamic properties and drying kinetics of *Bauhinia forficata* leaves. Nadi et al. [23] also reported that the thickness of an apple slice affected the values of effective moisture diffusivity, enthalpy, entropy, and Gibbs free energy. All the above studies imply that the thickness of kelp may be one of the most important factors affecting the drying kinetics and thermodynamic characteristics, and a major cause of non-uniform moisture content.

Therefore, this work aimed to shorten the drying time, on the premise of uniform moisture content, by studying the kinetics and thermodynamics of the heat pump drying of kelp. The effects of different temperatures, air velocities, and humidity on the drying characteristics of kelp with different thicknesses were explored. A mathematical model was built to reveal the moisture migration law and the thermodynamic properties' parameters were calculated to improve product quality with consistent moisture content. This work explains the reason for the difference in final moisture content, from a material nature point of view, and provides a theoretical basis for the thermodynamic calculation of a macroalgae heat pump drying system.

## 2. Materials and Methods

### 2.1. Materials

Fresh kelp was harvested from Xiapu County, Ningde. The thickness of each part was determined with a Vernier caliper and the initial moisture content was measured in an electric oven (BPJ-9123-a, Shanghai Instrument Manufacturing Co., Ltd., Shanghai, China), as shown in Table 1 (*n* = 8). Before drying, the kelp was cut carefully into 20 cm × 10 cm slices and divided into 33 groups with 10 slices in each.

**Table 1.** The thickness and moisture content of different parts of kelp.

| Specimen | Top Part | Main Part | Tail Part |
|---|---|---|---|
| Thickness (mm) | 3.5–4.2 | 2.0–3.5 | 0.8–2.0 |
| Moisture content (%) | 94.5 | 95.7 | 92.4 |

### 2.2. Heat Pump Dryer

A heat pump dryer (YCFZD-2HP, Hangzhou Emant Technology Co., Ltd.,Hangzhou, China) was used for the drying experiments (Figure 1). It consisted of a drying chamber (60 cm × 90 cm) and a heat pump system with control devices and a centrifugal air blower.

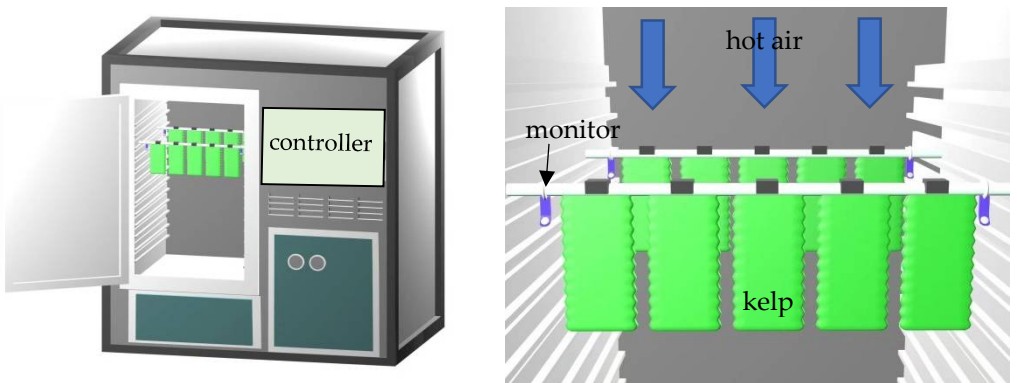

**Figure 1.** Schematic diagram of the heat pump drier and the geometric arrangement of kelp slices.

The temperature, air velocity, humidity, and thickness were taken as the four main influencing factors affecting kelp heat pump drying. In addition to the equipment temperature, humidity, and air velocity controller, 4 monitoring points were set around the kelp area in the chamber, and the parameters of different positions were measured by temperature–humidity recorders (DS1923, Shanghai Wodisen Electronic Technology Co., Ltd.,Shanghai, China) and a hand-held thermal anemometer (MODEL6004, Shenyang Jianye Max Instrument Co., Ltd., Shenyang, China). The temperature, humidity, and air velocity accuracy among the 4 monitoring points were within ±1.67 °C, ±2.97%, and ±0.04 m/s, respectively. All parameters were stable before the drying experiments.

### 2.3. Drying Procedure

The experiments were conducted according to Table 2. During the drying process, as shown in Figure 1, two rigid PVC pipes were suspended on the internal bracket, and the kelp slices were hung vertically in 2 rows of 5 slices using small stainless-steel clamps. The blower provided air from top to bottom through the kelp in parallel. The kelp weight was recorded hourly during the drying process using an electronic balance (ME4002E, METTLER TOLEDO, Shanghai, China). When the moisture content (wet base) fell to below 18%, the experiment was terminated. The total drying duration was 4–16 h, depending on the drying conditions and kelp thickness. The experiment was repeated 3 times for each group.

**Table 2.** Arrangement of heat pump drying of kelp.

| Group | Temperature (°C) | Air Velocity (m/s) | Humidity (%) | Thickness (mm) |
|:---:|:---:|:---:|:---:|:---:|
| 1 | 20 | 0.8 | 40 | |
| 2 | 30 | 0.8 | 40 | |
| 3 | 40 | 0.8 | 40 | |
| 4 | 50 | 0.8 | 40 | |
| 5 | 40 | 0.3 | 40 | 0.8–2.0 (A) |
| 6 | 40 | 0.8 | 40 | 2.0–3.5 (B) |
| 7 | 40 | 1.3 | 40 | 3.5–4.2 (C) |
| 8 | 40 | 0.3 | 20 | |
| 9 | 40 | 0.3 | 30 | |
| 10 | 40 | 0.3 | 40 | |
| 11 | 40 | 0.3 | 50 | |

*2.4. Moisture Content and Moisture Ratio*

Moisture content ($M_{db}$) was calculated as follows:

$$M_{wb} = \frac{m_t - m_0(1 - M_0)}{m_t} \times 100\% \tag{1}$$

$$M_{db} = \frac{M_{wb}}{1 - M_{wb}} \times 100\% \tag{2}$$

where $M_{wb}$ indicates the wet basis moisture content at a particular drying time $t$, g $H_2O$/g w.b.; $M_{db}$ indicates the dry basis moisture content at a particular drying time $t$, g $H_2O$/g d.b.; $m_t$ indicates the sample weight at a particular drying time $t$, g; $m_0$ indicates the initial weight of the sample, g; $M_0$ indicates the initial moisture content, g $H_2O$/g w.b.

The moisture ratio ($MR$) was calculated as follows:

$$MR = \frac{M_t - M_e}{M_o - M_e} \tag{3}$$

where $M_e$ indicates the equilibrium moisture content, g $H_2O$/g d.b.

Sappati et al. [24] measured the equilibrium moisture content ($M_e$) of kelp at temperatures of 40–70 °C and 25–50% humidity, and the $M_e$ value varied from 0.0707 to 0.1270 kg $H_2O$/kg dry solids, without any significant difference. Since the $M_e$ was far less than the $M_0$ and $M_t$, Formula (2) could be simplified as [25–28]

$$MR = \frac{M_t}{M_0} \tag{4}$$

*2.5. Mathematical Modeling of Drying Curves*

Henderson–Pabis, Page, and Lewis models are commonly used mathematic models of for most thin-layer drying of organic and biological materials [14,29]. These models are effective over the specific ranges of temperature, air velocity, and humidity for which they are developed. Linearization was conducted on each model (Table 3).

**Table 3.** Mathematical model of kelp heat pump drying.

| Model Name | Model | After Linearization | Reference |
|:---:|:---:|:---:|:---:|
| Henderson–Pabis | $MR = A\exp(-Kt)$ | $-\ln MR = -\ln A + Kt$ | [30] |
| Page model | $MR = \exp(-Kt^n)$ | $\ln[-\ln(MR)] = \ln K + n\ln t$ | [31] |
| Lewis model | $MR = \exp(-Kt)$ | $-\ln MR = Kt$ | [32] |

Note: $MR$ is the water ratio; $t$ is the drying time; $K$ is the change rate constant; $n$ is a shape parameter; $A$ is an undetermined rate coefficient.

### 2.6. Effective Moisture Diffusivity

Fick's unsteady second law equation was adopted [23,27], and the water diffusion of kelp in the heat pump drying process is given as follows:

$$MR = \frac{8}{\pi^2} \sum_{n=0}^{\infty} \frac{1}{(2n+1)^2} \exp\left(-\frac{(2n+1)^2 \pi^2 D_{eff} t}{L^2}\right) \tag{5}$$

where $D_{eff}$ is the effective moisture diffusivity of the material, $m^2/s$; $L$ is the material thickness of the kelp, m; $n$ is the positive integer; $t$ is the drying time, s. Page [31] approximated the $\frac{8}{\pi^2}$ ratio as being equal to unity, and $n$ is taken as 1 for longer drying times [27,33]. Thus, the equation was further simplified to the first term of the series and expressed in logarithmic form as follows:

$$\ln MR = \ln\left(\frac{8}{\pi^2}\right) - \left(\frac{\pi^2 D_{eff}}{L^2} t\right) \tag{6}$$

### 2.7. Thermodynamic Property Parameters

Drying activation energy $E_a$ is calculated according to the formula

$$D_{eff} = D_0 \exp\left[-\frac{E_a}{R(T + 273.15)}\right] \tag{7}$$

where $D_0$ indicates the diffusion base in the material, $m^2/s$; $E_a$ indicates the drying activation energy of the material, J/mol; $R$ is the molar constant of gas, whose value is 8.314 J/(mol·K); $T$ is the drying temperature of kelp, °C.

The drying rate constant $k$ ($s^{-1}$) can be expressed as [23]

$$k = \left(\frac{k_B T_{abs}}{h_p}\right) \exp\left(\frac{\Delta S}{R}\right) \exp\left(\frac{-\Delta H}{R T_{abs}}\right) \tag{8}$$

Transferred into logarithmic form, it is

$$\Delta S = R\left[\ln D_0 - \ln\left(\frac{k_B}{h_p}\right) - \ln T_{abs}\right] \tag{9}$$

$$\Delta H = E_a - R T_{abs} \tag{10}$$

$$\Delta G = \Delta H - T_{abs} \Delta S \tag{11}$$

where $\Delta H$ is the enthalpy change, J/mol; $\Delta S$ is the entropy change, J/mol·K; $\Delta G$ is the Gibbs free energy change, J/mol; $k_B$ is the Boltzmann constant ($1.380 \times 10^{-23}$ J/K), $h_p$ is the Planck's constant ($6.626 \times 10^{-34}$ J·s), $R$ is the universal gas constant (8.314 J/mol·K), and $T_{abs}$ is the absolute temperature (K).

### 2.8. Statistical Analyses

Data were analyzed using IBM SPSS Statistics 22 (SPSS Lnc., Chicago, IL, USA) and presented as mean values from triplicate samples. The coefficient of determination ($R^2$) of the model was calculated via the following equations [19].

$$R^2 = 1 - \frac{\sum_{i=1}^{n}\left(MR_{pre,i} - MR_{exp,i}\right)^2}{\sum_{i=1}^{n}\left(\overline{MR}_{pre,i} - MR_{exp,i}\right)^2} \tag{12}$$

where $MR_{pre,i}$ and $MR_{exp,i}$ represent the predicted value and the experimental value, respectively; and $n$ is the number of data.

The residual distribution was evaluated to check the randomness between predicted values and the experimental data.

## 3. Results

### 3.1. Effect of Heat Pump Drying on Dehydration Characteristics

3.1.1. Effect of Temperature on Drying Characteristics

The change in the moisture ratio (*MR*) and drying rate under different drying temperatures and kelp thickness is shown in Figure 2. The moisture content reduced to 18% within 4–16 h, and increasing the drying temperature and decreasing the kelp thickness could shorten the drying time. The drying times of the 30 °C, 40 °C, and 50 °C groups were reduced by 22.5%, 39.2%, and 55.6%, respectively, compared with the 20 °C group. The average drying time with a thickness of 0.8–2 mm was 26.5%, 41.9% shorter than that of 2–3.5 mm and 3.5–4.2 mm, respectively. This is consistent with Chen et al. [34], who found that drying temperature and kelp extract thickness could significantly affect the drying rate by 40–50%.

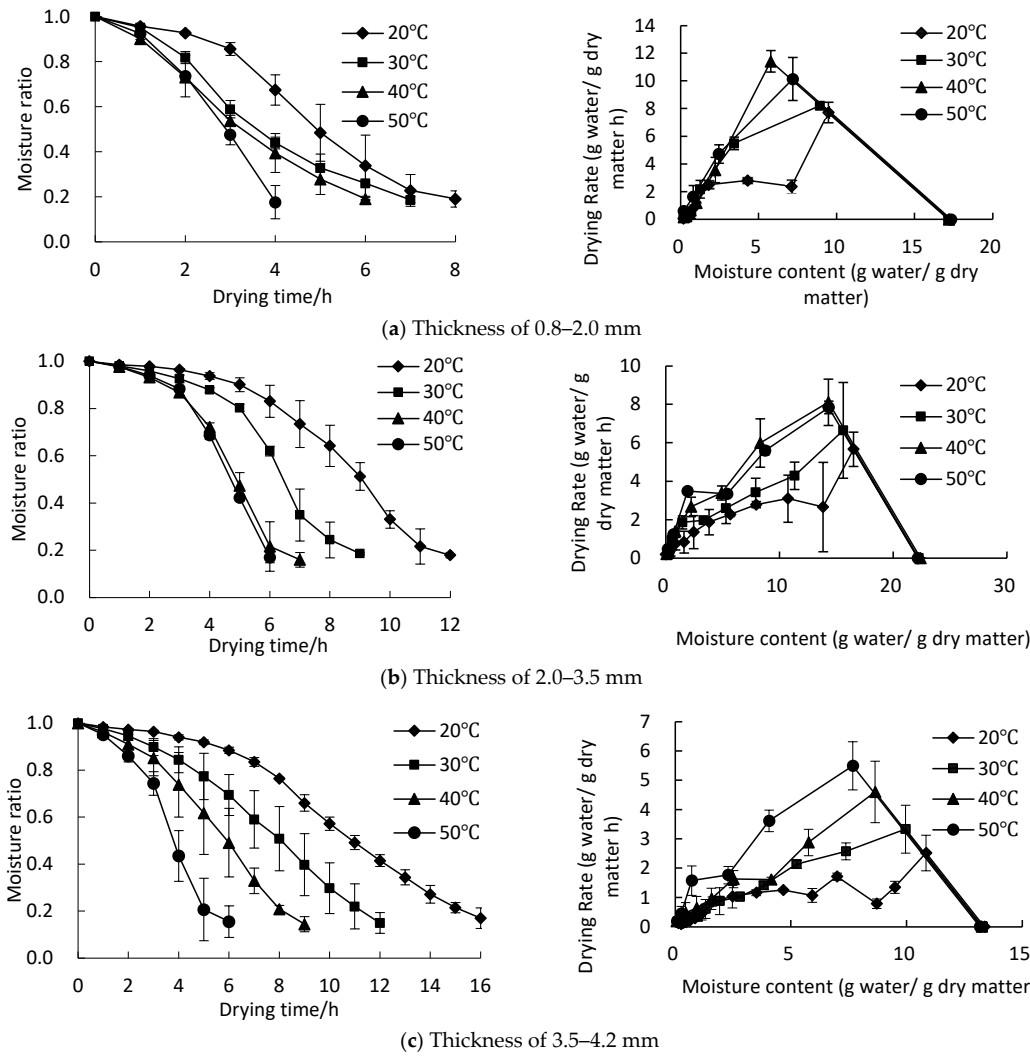

(**a**) Thickness of 0.8–2.0 mm

(**b**) Thickness of 2.0–3.5 mm

(**c**) Thickness of 3.5–4.2 mm

**Figure 2.** Drying curves of kelp under different drying temperatures and thicknesses.

The kelp moisture ratio decreased with the extension of drying time, while the drying rate slowed down in the latter part of the drying period. This is because the parenchyma cells and spaces between the cells were fully filled with moisture in the early stages, and cells could stretch freely, allowing moisture to escape through the loose passageway. Although there was moisture transfer resistance inside the material at this time, the drying

rate was mainly affected by material surface evaporation [35], but, along with the heat pump drying, moisture within the kelp declined significantly. The drying limiting factor gradually changed from being surface evaporation control to internal diffusion control. Of course, this transition was related to many factors. From Figure 2, both temperature and thickness had an impact on this transition. The decelerated drying stage was obviously longer with thin kelp leaves and lower temperatures. This is because the thin leaves contained less moisture and the low-temperature group offered a smaller temperature gradient, which prolonged the decelerated drying process.

### 3.1.2. Effect of Air Velocity on Drying Characteristics

Figure 3 shows the change in moisture ratio (*MR*) and drying rate under different drying air velocities and thicknesses. It can be seen from Figure 3 that the drying time was shorter with an increase in air velocity and decrease in thickness. The average kelp drying time in 3.5–4.2 mm and 2.0–3.5 mm was 47.4%, 37.0% longer than that of 0.8–2.0 mm. When the air velocity was 1.3 m/s, the kelp drying time reduced by 50.0% (0.8–2.0 mm), 64.3% (2.0–3.5 mm), and 53.8% (3.5–4.2 mm), respectively, compared with a 0.3 m/s air velocity. As the air velocity increased, the kelp surface heat and mass transfer improved, and the water vapor escaping from the materials was rapidly removed and replaced by fresh dry air, accelerating the surface moisture evaporation and shortening the drying time. This was consistent with the findings of both the hot air drying of yam slices [33] and of pepper leaves [29].

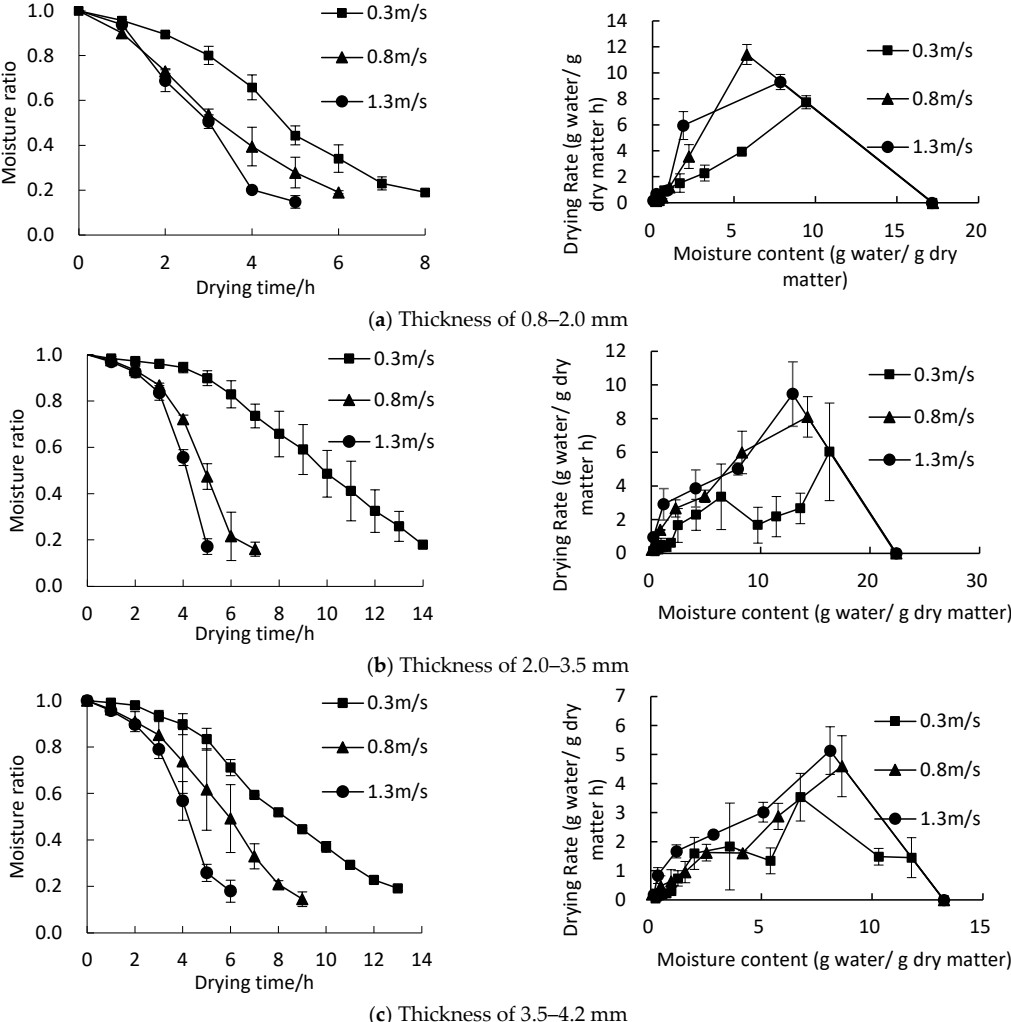

**Figure 3.** Drying curves of kelp under different drying air velocities and thicknesses.

### 3.1.3. Effect of Humidity on Drying Characteristics

Figure 4 illustrates the change in the moisture ratio (*MR*) and drying rate under different drying humidity levels and thicknesses. It can be seen that when the drying temperature, air velocity, and thickness are constant, a reduction in dry humidity shortens the drying time. When the humidity was reduced from 50% to 20%, the kelp drying time was shortened by 41.0% on average. This is similar to the findings of a study by Xu et al. [36] on the hot air drying of finger citron slices. When the humidity was 40% and 50%, the final wet base moisture content of kelp (3.5–4.2 mm) terminated at 19.2% and 19.7%, respectively, and prolonged drying could not remove any more moisture. This is consistent with the conclusion drawn by Namkanisorn and Murathathunyaluk [37] in the equilibrium moisture content of galangal. This is due to the intracellular and entrapped water dominating in the latter drying period, which is already difficult to migrate outwards, particularly at low humidity gradients.

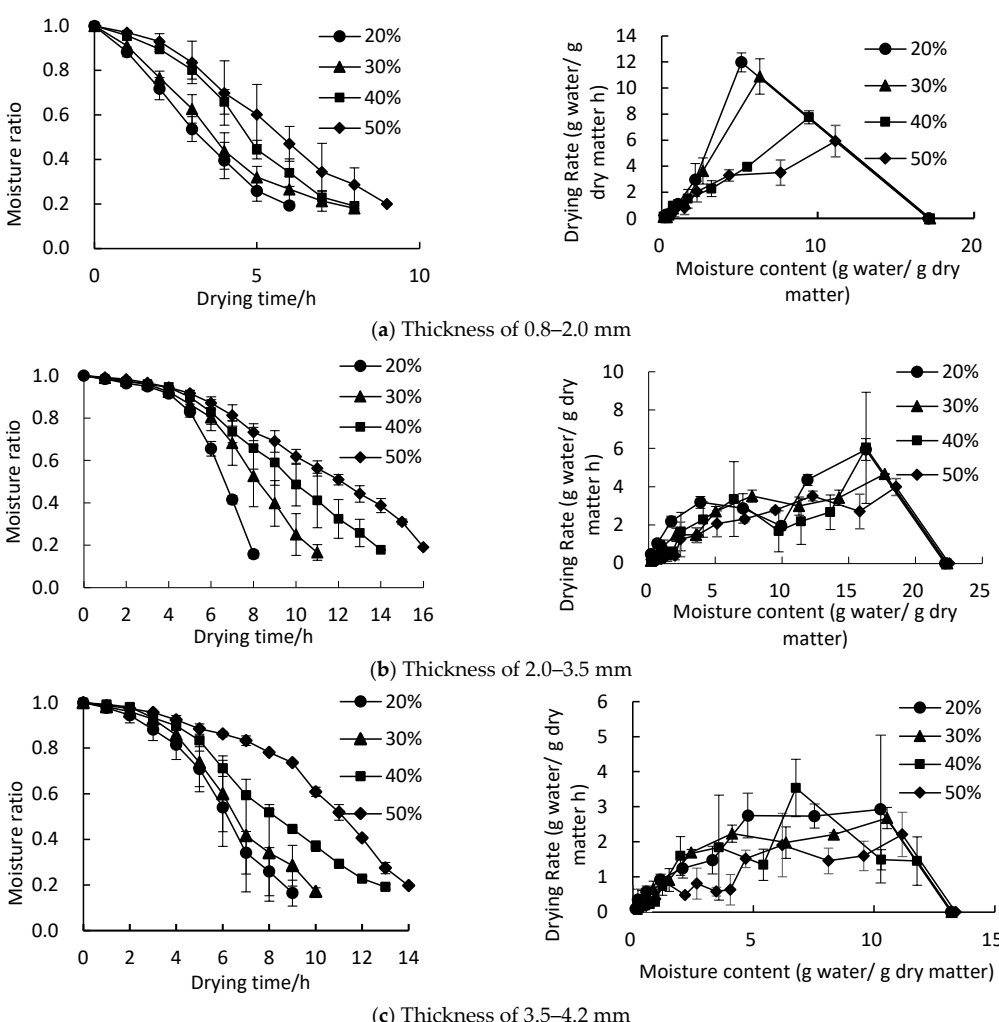

**Figure 4.** Drying curves of kelp under different drying humidity levels and thicknesses.

### 3.1.4. Effect of Kelp Thickness on Drying Time

Figure 5 shows the kelp drying time under different drying conditions and thicknesses. The difference in the drying time caused by thickness was between 2 h and 10 h, with a full drying duration of 4–16 h. In the T20, V0.3, and H50 groups, the kelp drying time varied dramatically with different thicknesses. The top part (3.5–4.2 mm) took 10 h, 9 h, and 8 h, respectively, longer than the tail part (0.8–2.0 mm). This indicates that the moisture content of the whole kelp varied greatly after the same drying time under such conditions. When the kelp moisture content was below 18%, the edges could become over-dried, while there

was still excessive moisture in the top part. By contrast, the effect of thickness on drying time was less obvious under higher temperatures, air velocity, and lower humidity. This phenomenon was similar to that observed by Tham et al. [25], who reported that drying air at a high temperature and low humidity can reduce the inhomogeneity of air temperature and humidity and improve drying efficiency. The optimal values for the heat pump drying process were determined as a temperature of 40 °C, air velocity of 1.3 m/s, and humidity of 40%. The moisture content could reach 18% within 5 h with improved uniformity in moisture content. This corresponds with the conclusion drawn by Hu et al. [11], who claimed that the optimum temperature for kelp knot heat pump drying was 45 °C, with a drying time of 260 min.

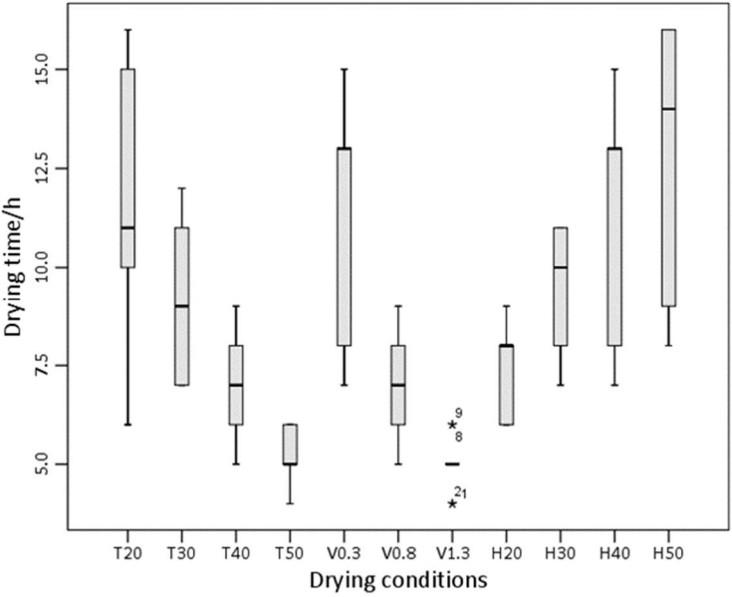

**Figure 5.** Drying time of kelp with different thicknesses under different drying conditions (T: temperature; V: air velocity; H: humidity; *: extremum value).

### 3.2. Determination of Drying Model

Heat pump drying is a complex heat and mass transfer drying process. The experimental data of kelp drying at different temperatures, air velocities, humidity, and thicknesses were represented by the moisture ratio and fitted to three models (Table 4). The correlation coefficient of the Page model was 0.9663–0.9993, and the high fitting degree indicated that the Page model can accurately describe the drying kinetics of kelp.

**Table 4.** Curve-fitting results under different drying conditions.

| No. | $\ln MR$—$t$ | | | $\ln[-\ln(MR)]$—$\ln t$ | | |
|---|---|---|---|---|---|---|
| | $\ln A$ | $K$ | $R^2$ | $\ln K$ | $n$ | $R^2$ |
| 1A | 0.4672 | 0.2096 | 0.9479 | −4.283 | 2.381 | 0.966 |
| 1B | 0.4662 | 0.1313 | 0.8826 | −6.137 | 2.618 | 0.945 |
| 1C | 0.4353 | 0.1061 | 0.8749 | −5.711 | 2.209 | 0.964 |
| 2A | 0.4124 | 0.2487 | 0.9837 | −2.591 | 1.646 | 0.977 |
| 2B | 0.5470 | 0.1867 | 0.8537 | −5.378 | 2.648 | 0.959 |
| 2C | 0.4214 | 0.1647 | 0.9128 | −4.433 | 1.983 | 0.994 |
| 3A | 0.3928 | 0.2635 | 0.9857 | −2.170 | 1.503 | 0.993 |
| 3B | 0.5583 | 0.2353 | 0.8132 | −4.797 | 2.790 | 0.958 |
| 3C | 0.4750 | 0.1934 | 0.8849 | −3.991 | 2.085 | 0.987 |
| 4A | 0.6750 | 0.4139 | 0.8849 | −2.941 | 2.484 | 0.995 |
| 4B | 0.8562 | 0.2614 | 0.7441 | −5.176 | 3.110 | 0.948 |
| 4C | 0.5998 | 0.3112 | 0.8842 | −3.704 | 2.476 | 0.978 |

**Table 4.** *Cont.*

| No. | lnMR—t | | | ln[−ln(MR)]—lnt | | |
|---|---|---|---|---|---|---|
| | ln*A* | *K* | $R^2$ | ln*K* | *n* | $R^2$ |
| 5A | 0.4833 | 0.2252 | 0.9334 | −3.666 | 2.059 | 0.989 |
| 5B | 0.4079 | 0.1111 | 0.8973 | −5.658 | 2.295 | 0.956 |
| 5C | 0.4740 | 0.1418 | 0.9288 | -5.399 | 2.350 | 0.989 |
| 6A | 0.3928 | 0.2635 | 0.9857 | −2.170 | 1.503 | 0.993 |
| 6B | 0.5583 | 0.2353 | 0.8132 | −4.797 | 2.790 | 0.958 |
| 6C | 0.4750 | 0.1934 | 0.8849 | −3.991 | 2.085 | 0.987 |
| 7A | 0.6824 | 0.4135 | 0.9216 | −2.330 | 1.894 | 0.984 |
| 7B | 0.6204 | 0.3023 | 0.6865 | −5.104 | 3.380 | 0.921 |
| 7C | 0.6077 | 0.2937 | 0.8461 | −4.185 | 2.664 | 0.967 |
| 8A | 0.4315 | 0.2850 | 0.977 | −2.112 | 1.475 | 0.999 |
| 8B | 0.5719 | 0.1979 | 0.6182 | −5.849 | 2.849 | 0.885 |
| 8C | 0.5268 | 0.1978 | 0.8614 | −4.658 | 2.356 | 0.984 |
| 9A | 0.3339 | 0.2230 | 0.985 | −2.219 | 1.381 | 0.989 |
| 9B | 0.5192 | 0.1543 | 0.7807 | −5.993 | 2.653 | 0.955 |
| 9C | 0.4818 | 0.1710 | 0.8853 | −5.105 | 2.461 | 0.984 |
| 10A | 0.4833 | 0.2252 | 0.9334 | −3.666 | 2.059 | 0.989 |
| 10B | 0.4079 | 0.1111 | 0.8973 | −5.658 | 2.295 | 0.956 |
| 10C | 0.4740 | 0.1418 | 0.9288 | −5.399 | 2.350 | 0.989 |
| 11A | 0.4176 | 0.1808 | 0.9366 | −3.964 | 2.043 | 0.995 |
| 11B | 0.3440 | 0.0873 | 0.8544 | −5.787 | 2.187 | 0.986 |
| 11C | 0.3865 | 0.0985 | 0.7726 | −5.428 | 2.084 | 0.976 |

### 3.3. Verification of Kinetic Model

The fitting accuracy of the mathematical drying model and experimental data was further evaluated by residual analysis (Figure 6) and verified (Figure 7). Figure 6 depicts the residual graph of the Page (A) and Henderson–Pabis (B) models for all groups (total experimental points = 248). The residual distribution for the Page model varied from −0.246 to 0.165, while for the Henderson–Pabis model, the residual values ranged from −0.262 to 0.736 in a U pattern, indicating that the Page model represents more accurately the drying mechanism in comparison with the Henderson–Pabis model.

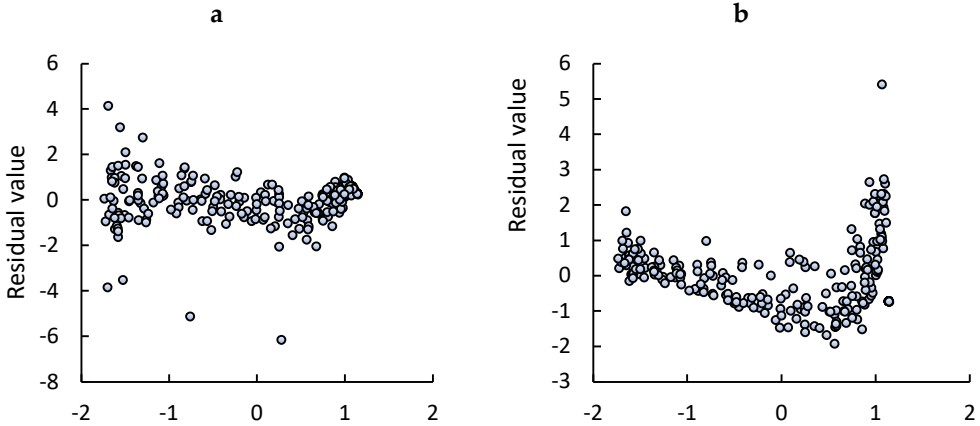

**Figure 6.** Residual plot distributions of (**a**) Page model, (**b**) Henderson-Pabis model for all groups.

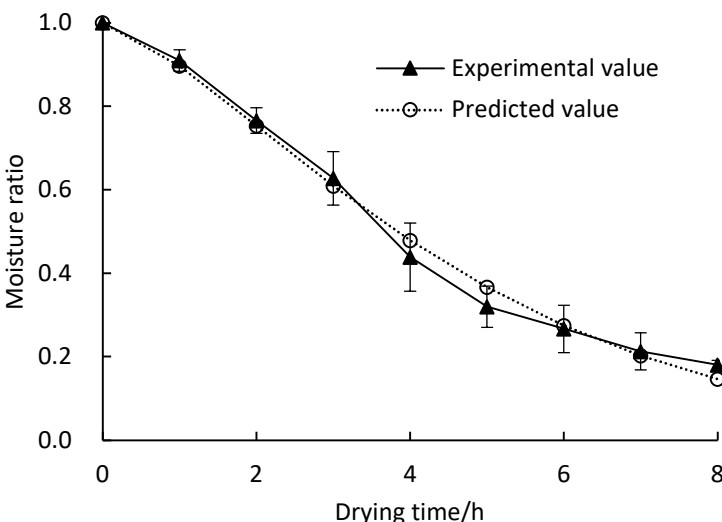

**Figure 7.** Page model validation.

The predicted and experimental values were compared at 40 °C, 0.3 m/s, and 30%, as shown in Figure 7. The actual value curve of the test and the predicted value curve of the Page model fit well. Similar findings were reported by Singhanat [38], Hao et al. [39], and Almeida et al. [21], who also found that the Page model was most effective in describing the heat pump drying characteristics of ginger, lemon slices, and Achachairu peels. However, the Page model does not fit all the products; for instance, the Midilli and Kucuk model was selected to represent the drying kinetics of banana slices [14] and yacon slices [27], while *B. forficata* link leaves were described by the Valcam model [22].

### 3.4. Effective Moisture Diffusivity

The effective moisture diffusivity ($D_{eff}$) was calculated (Table 5) in order to explore the moisture transfer characteristics of kelp in heat pump drying. It showed obviously that the $D_{eff}$ of heat-pump-dried kelp increased from $4.4307 \times 10^{-11}$ to $12.9957 \times 10^{-11}$ m$^2$/s when the temperature rose from 20 to 50 °C (3.5–4.2 mm). When the air velocity rose from 0.3 to 1.3 m/s, the $D_{eff}$ increased from $5.9216 \times 10^{-11}$ to $12.2649 \times 10^{-11}$ m$^2$/s (3.5–4.2 mm). When the humidity was increased from 20% to 50%, the $D_{eff}$ reduced from $8.2601 \times 10^{-11}$ to $4.1134 \times 10^{-11}$ m$^2$/s (3.5–4.2 mm). When the thickness was increased from 0.8–2 mm to 3.5–4.2 mm, the $D_{eff}$ was improved from $1.2314 \times 10^{-11}$ to $7.1409 \times 10^{-11}$ m$^2$/s (40 °C, 30%, 0.3 m/s). This indicates that the temperature, air velocity, and thickness exert positive effects on the $D_{eff}$ value; conversely, humidity exerts a negative effect on the $D_{eff}$ increase. This is consistent with the findings of *Bauhinia forficata* link leaves [22], apple slices [23], and pepper leaves [29]. By contrast, the $D_{eff}$ of kelp is lower than that of Achachairu peels [21] and banana slices [14], but higher than that of *Bauhinia forficata* link leaves [22] and *Piper umbellatum L.* leaves [29] at the same drying temperature. The $D_{eff}$ value was related to the physicochemical properties and moisture content, as well as temperature. On the one hand, the kelp initial moisture content was up to 92.4–95.7%, which contributed to the $D_{eff}$ value. However, on the other hand, the kelp surface was compact and leathery, retarding the free diffusion of moisture. In addition, kelp is rich in dietary fiber, which has a strong water-holding capacity and inhibits the migration of water molecules.

**Table 5.** Effective moisture diffusivity under different drying conditions.

| No. | Linear Regression Fitting Formula | $D_{eff}$ $(10^{-11}\ m^2 \cdot s^{-1})$ | $R^2$ | No. | Linear Regression Fitting Formula | $D_{eff}$ $(10^{-11}\ m^2 \cdot s^{-1})$ | $R^2$ |
|---|---|---|---|---|---|---|---|
| 1A | $\ln MR = -5.822 \times 10^{-5}t + 0.4672$ | 1.1574 | 0.9479 | 7A | $\ln MR = -1.148 \times 10^{-4}t + 0.6824$ | 2.2833 | 0.9216 |
| 1B | $\ln MR = -3.647 \times 10^{-5}t + 0.4662$ | 2.7975 | 0.8826 | 7B | $\ln MR = -8.397 \times 10^{-5}t + 0.6204$ | 6.4408 | 0.6865 |
| 1C | $\ln MR = -2.947 \times 10^{-5}t + 0.4353$ | 4.4307 | 0.8749 | 7C | $\ln MR = -8.158 \times 10^{-5}t + 0.6077$ | 12.2649 | 0.8461 |
| 2A | $\ln MR = -6.908 \times 10^{-5}t + 0.4124$ | 1.3733 | 0.9837 | 8A | $\ln MR = -7.916 \times 10^{-5}t + 0.4315$ | 1.5738 | 0.977 |
| 2B | $\ln MR = -5.186 \times 10^{-5}t + 0.547$ | 3.9778 | 0.8537 | 8B | $\ln MR = -5.497 \times 10^{-5}t + 0.5719$ | 4.2165 | 0.6182 |
| 2C | $\ln MR = -4.575 \times 10^{-5}t + 0.4214$ | 6.8779 | 0.9128 | 8C | $\ln MR = -5.494 \times 10^{-5}t + 0.5268$ | 8.2601 | 0.8614 |
| 3A | $\ln MR = -7.319 \times 10^{-5}t + 0.3928$ | 1.4550 | 0.9857 | 9A | $\ln MR = -6.194 \times 10^{-5}t + 0.3339$ | 1.2314 | 0.985 |
| 3B | $\ln MR = -6.536 \times 10^{-5}t + 0.5583$ | 5.0133 | 0.8132 | 9B | $\ln MR = -4.286 \times 10^{-5}t + 0.5192$ | 3.2875 | 0.7807 |
| 3C | $\ln MR = -5.972 \times 10^{-5}t + 0.4750$ | 8.0764 | 0.8849 | 9C | $\ln MR = -4.75 \times 10^{-5}t + 0.4818$ | 7.1409 | 0.8853 |
| 4A | $\ln MR = -1.149 \times 10^{-4}t + 0.675$ | 2.2855 | 0.8449 | 10A | $\ln MR = -6.255 \times 10^{-5}t + 0.4833$ | 1.2435 | 0.9334 |
| 4B | $\ln MR = -7.261 \times 10^{-5}t + 0.5862$ | 5.5694 | 0.7441 | 10B | $\ln MR = -3.086 \times 10^{-5}t + 0.4079$ | 2.3671 | 0.8973 |
| 4C | $\ln MR = -8.644 \times 10^{-5}t + 0.5998$ | 12.9957 | 0.8842 | 10C | $\ln MR = -3.938 \times 10^{-5}t + 0.4740$ | 5.9216 | 0.9288 |
| 5A | $\ln MR = -6.255 \times 10^{-5}t + 0.4833$ | 1.2435 | 0.9334 | 11A | $\ln MR = -5.022 \times 10^{-5}t + 0.4176$ | 0.9984 | 0.9366 |
| 5B | $\ln MR = -3.086 \times 10^{-5}t + 0.4079$ | 2.3671 | 0.8973 | 11B | $\ln MR = -2.425 \times 10^{-5}t + 0.3440$ | 1.8600 | 0.8544 |
| 5C | $\ln MR = -3.938 \times 10^{-5}t + 0.4740$ | 5.9216 | 0.9288 | 11C | $\ln MR = -2.736 \times 10^{-5}t + 0.3865$ | 4.1134 | 0.7726 |
| 6A | $\ln MR = -7.319 \times 10^{-5}t + 0.3928$ | 1.4550 | 0.9857 | | | | |
| 6B | $\ln MR = -6.536 \times 10^{-5}t + 0.5583$ | 5.0133 | 0.8132 | | | | |
| 6C | $\ln MR = -5.372 \times 10^{-5}t + 0.4750$ | 8.0764 | 0.8849 | | | | |

### 3.5. Thermodynamic Parameters of Kelp with Different Thicknesses

The thermodynamic parameters of kelp were calculated and are shown in Table 6.

**Table 6.** Thermodynamic parameters of kelp with different thicknesses.

| Thickness | Drying Conditions | $E_a$ (kJ/mol) | $\Delta H$ (kJ/mol) | $\Delta S$ (J/mol·K) | $\Delta G$ (kJ/mol) |
|---|---|---|---|---|---|
| 0.8–2.0 mm | 20 °C | 16.38 | 13.94 | −398.26 | 130.69 |
| | 30 °C | | 13.86 | −398.96 | 134.8 |
| | 40 °C | | 13.78 | −400.47 | 139.18 |
| | 50 °C | | 13.69 | −398.6 | 142.5 |
| | 0.3 m/s | | 13.78 | −401.78 | 139.59 |
| | 0.8 m/s | | 13.78 | −400.47 | 139.18 |
| | 1.3 m/s | | 13.78 | −396.72 | 138.01 |
| | 20% | | 13.78 | −399.82 | 138.98 |
| | 30% | | 13.78 | −401.86 | 139.62 |
| | 40% | | 13.78 | −401.78 | 139.59 |
| | 50% | | 13.78 | −403.6 | 140.16 |
| 2.0–3.5 mm | 20 °C | 18.21 | 15.77 | −384.68 | 128.54 |
| | 30 °C | | 15.69 | −384.08 | 132.12 |
| | 40 °C | | 15.61 | −384.34 | 135.96 |
| | 50 °C | | 15.52 | −385.53 | 140.11 |
| | 0.3 m/s | | 15.61 | −390.58 | 137.92 |
| | 0.8 m/s | | 15.61 | −384.34 | 135.96 |
| | 1.3 m/s | | 15.61 | −382.26 | 135.31 |
| | 20% | | 15.61 | −385.78 | 136.41 |
| | 30% | | 15.61 | −387.85 | 137.06 |
| | 40% | | 15.61 | −390.58 | 137.92 |
| | 50% | | 15.61 | −392.58 | 138.54 |
| 3.5–4.2 mm | 20 °C | 26.66 | 24.22 | −352.03 | 127.42 |
| | 30 °C | | 24.14 | −351.65 | 130.74 |
| | 40 °C | | 24.06 | −353.39 | 134.72 |
| | 50 °C | | 23.97 | −352.33 | 137.83 |
| | 0.3 m/s | | 24.06 | −355.97 | 135.53 |
| | 0.8 m/s | | 24.06 | −353.39 | 134.72 |
| | 1.3 m/s | | 24.06 | −349.92 | 133.63 |
| | 20% | | 24.06 | −353.21 | 134.66 |
| | 30% | | 24.06 | −354.42 | 135.04 |
| | 40% | | 24.06 | −355.97 | 135.53 |
| | 50% | | 24.06 | −359 | 136.48 |

The activation energy ($E_a$) is an important index to reflect the moisture bonding ability of materials, indicating the energy required for water molecules to change from a normal to an active state that is prone to dehydration. The $E_a$ of kelp with a thickness of 0.8–2.0 mm, 2.0–3.5 mm, and 3.5–4.2 mm were calculated to be 16.38 kJ/mol, 18.21 kJ/mol, and 26.66 kJ/mol, respectively. The $E_a$ values of various food materials range from 12.7 to 110 kJ/mol [40]. The energy threshold of the top part was 62.76% higher than that of the tail part, indicating that tail parts can start the dehydration process with lower energy supplied in the initial heating-up period. The $E_a$ difference was in line with the effective moisture diffusivity of kelp with different thicknesses, and this is the intrinsic reason for the faster drying of thinner kelp leaves.

The enthalpy ($\Delta H$) in the drying process represents the energy required by moisture to be removed from kelp. The $\Delta H$ values were positive and decreased with drying temperature, which illustrates that the drying process is endothermic and requires less energy at higher drying temperatures. This is consistent with Costa et al. [41] on the thermodynamic characteristics of fruit peel drying at different temperatures. Compared with the drying temperature, the thickness of kelp exhibited a more obvious effect on the $\Delta H$. The $\Delta H$ of kelp with a thickness of 3.5–4.2 mm was 174.54% and 154.09% for the $\Delta H$ of 0.8–2.0mm and 2.0–3.5 mm, respectively. The results showed that the energy required by the top section of the kelp was 54.09%–74.54% higher than in other parts of kelp.

Entropy plays a definitive role in the chaos of the materials, the change in energy, and the stability of the process. The results showed that the entropy decreased significantly during kelp drying, signifying that heat pump drying improved the intermolecular order of the kelp, which stabilized the dried kelp tissue.. In addition, both temperature and air velocity exerted negative effects on the $\Delta S$ value, but thickness had a positive effect. Similar results were found with apple slices [23], but the results with *Bauhinia forficata* link leaves differed [22].

The Gibbs free energy ($\Delta G$) assesses the spontaneity of moisture desorption [42] and provides a better view on the thermodynamic driving forces influencing reactions [23]. When the phenomenon occurs spontaneously, $\Delta G$ is negative, meaning that no external energy is required, while the results showed that $\Delta G$ remained positive under all the investigated conditions, meaning that the process did not occur spontaneously. This corresponds with the $\Delta H$ result, suggesting that kelp drying is an endergonic process that requires energy from the environment for the reaction to occur. Furthermore, the $\Delta G$ value was increased with drying temperature and thickness, which is consistent with studies by Nadi et al. in terms of apple slice drying [23] and Corrêa et al. in terms of coffee drying [42]. However, the results differed from Costa et al. [41] and Silva et al. [22], who found that the $\Delta G$ decreased with increments in the drying temperature and thickness of the materials, respectively. These findings reflect the complex characteristics of internal energy reactions.

## 4. Conclusions

This study elucidated the kinetics and thermodynamics of heat pump drying for kelp, and mapped the apparent drying non-uniformity with the inner mechanism of heat and mass transfer, which gave an insight into the drying efficiency and quality improvement.

The temperature, air velocity, humidity, and thickness affected the kinetic characteristics of heat pump drying for kelp. With the increase in temperature and air velocity and a decrease in humidity, the effect of thickness on moisture uniformity reduced.

The Page model was established to predict the moisture migration principle of kelp. The kelp $D_{eff}$ was calculated as $1.00 \times 10^{-11}$–$13.00 \times 10^{-11}$ m²/s, increasing with drying temperature, air velocity, and thickness but decreasing with humidity.

The thermodynamic parameters of kelp heat pump drying were calculated. The kelp thickness had a significant effect on the thermodynamic characteristics. The kelp $E_a$ was 16.38–26.66 kJ/mol. Meanwhile, $\Delta H$ was 13.69–24.22 kJ/mol, $\Delta S$ was $-403.60$–$-349.92$ J/(mol·K), and $\Delta G$ was 127.42–142.50 kJ/mol. These results provide a theoretical basis for the ther-

modynamic calculation of a heat pump drying system for kelp and for improvements in macroalgae drying technology.

To shorten the drying time on the premise of uniformity in moisture content, the optimal drying parameters were determined as a temperature of 40 °C, air velocity of 1.3 m/s, and humidity of 40%.

**Author Contributions:** Conceptualization, Q.Z., G.Z. and S.X.; methodology, Q.Z. and M.Z.; software, S.L.; validation, M.Z.; resources, G.Z.; writing—original draft preparation, Q.Z.; writing—review and editing, G.M., X.L. and S.X.; project administration, G.Z. All authors have read and agreed to the published version of the manuscript.

**Funding:** This research was funded by National Key R&D Program of China (2019YFD0901800); Education Department of Liaoning Province (JL202011); Ocean Research Center of Liaoning Province (DL201908); Ocean and Fisheries Department of Liaoning Province (201722); Key Laboratory of Environment Controlled Aquaculture (Dalian Ocean University) Ministry of Education (202203); National Key R&D Program of China (2020YFD0900600); Natural Science Foundation of Liaoning Province (2020-BS-214).

**Institutional Review Board Statement:** Not applicable.

**Informed Consent Statement:** Not applicable.

**Data Availability Statement:** All data that support the findings of this study are included within the article.

**Conflicts of Interest:** The authors declare no conflict of interest.

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
