# Peer review of "Heat Pump Drying of Kelp (Laminaria japonica): Drying Kinetics and Thermodynamic Properties"

_processes, doi:10.3390/pr10030514_

Round 1

Reviewer 1 Report

The title of the manuscript does correspond to its content.  The manuscript is of scientific importance as it shows the effect of drying conditions (temperature, air velocity, humidity) and thickness of the kelp slices using a heat pump. The heat pump used to dry food is still a new topic. Therefore, in my opinion, the introduction should be supplemented with the construction of this device.

L135 and 136: unit, must be (J/mol·K)

The calculated thermodynamic parameters prove that the drying process does not take place spontaneously. What could be the reason for this result? (L315-320)

Overall, the quality of the research work is well, but revisions are recommended. Some revisions are highlighted in the manuscript.

Author Response

Dear reviewer,

Thank you so much for taking your time to review this manuscript. I appreciate all your comments and suggestions! Please find my itemized responses in below and my revisions in the re-submitted files.

Point 1. The title of the manuscript does correspond to its content. The manuscript is of scientific importance as it shows the effect of drying conditions (temperature, air velocity, humidity) and thickness of the kelp slices using a heat pump. The heat pump used to dry food is still a new topic. Therefore, in my opinion, the introduction should be supplemented with the construction of this device.

Response 1. The construction and working principle of the heat pump device was supplemented in the introduction (paragraph 2), and the accuracy and dimension was supplemented in materials and methods (2.2 Heat pump dryer).

Point 2. L135 and 136: unit, must be (J/mol·K)

Response 2. It was revised to be (J/mol·K).

Point 3. The calculated thermodynamic parameters prove that the drying process does not take place spontaneously. What could be the reason for this result? (L315-320)

Response 3. When the phenomenon occurs spontaneously, â–³G is negative, meaning that no external energy is required. While the results showed that â–³G was positive under all the investigated conditions, meaning that external energy was necessary for the drying process to take place. This was corresponding with the result of â–³H, suggesting that drying of kelp is an endergonic process that requires energy from the environment for the reaction to occur. This was supplemented in the manuscript (3.5 Thermodynamic parameters of kelp in different thicknesses).

Point 4. Overall, the quality of the research work is well, but revisions are recommended. Some revisions are highlighted in the manuscript.

Response 4. Thank you for the highlighted revisions, all the mistakes was corrected, including adding the space, correcting the unit and the reference format, etc.

Thanks again!

Best regards,

Qian Zhang

Feb 24, 2022

Reviewer 2 Report

The submitted manuscript entitled “Heat pump drying of kelp (Laminaria japonnica): drying kinetics and thermodynamic properties” dealt with the heat pump drying analysis regarding kelp by fitting the measured data related drying performance and thermodynamics to proper mathematical model for speculating a possible optimal condition for kelp with the aid of a heat pump.  The authors addressed strong and reliable justifications for their objective and provide solid research results with sufficient discussion. It is generally well-written with accordance to the field the special issue. I would recommend the manuscript to accepted in its present form.

Some minor revision suggestions:

Please recheck on all the abbreviated terms for equations, e.g., line 104-105 m_0 for initial sample weight or anywhere else.

English can still be improved.

Author Response

Dear reviewer,

Thank you so much for taking your time to review this manuscript. I appreciate all your comments and suggestions! Please find my itemized responses in below and my revisions in the re-submitted files.

Point 1. Please recheck on all the abbreviated terms for equations, e.g., line 104-105 m_0 for initial sample weight or anywhere else.

Response 1. The mistakes were corrected. The equations were rechecked, including on all the abbreviated terms and units.

Point 2. English can still be improved.

Response 2. English writing was improved on words and phrases expressions.

Thanks again!

Best regards,

Qian Zhang

Feb 24, 2022

Reviewer 3 Report

Although the paper gives same interesting experimental result on drying of kelp, due to necessary improvements in methodology an analysis, it can not be published in the present form.

It is necessary to describe in more detail the drying arrangement and method: What can be mentioned about the stability (in time) and homogenety (in space) of the air temperature, the air humidity and velocity during the experiments . What is the geometric arrangement oft he slices, ist he air flowing parallely or is it a cross-flow arrangement….?

The equation (1) seems to be wrong. Either it must be divided  by m0 (then it would be on a wet basis) or by the dry mass.

In equation (2) and ist explanation it stays unclear, what equilibrium moisture content is used. The equilibrium moisture content is dependent on the air humidity and temperature!

Equation (4) does not fit to equation (3) since for t=0 equation (4) would not be equal to 1!

In equation (5) and ist explanation there is a misinterpretation. Either T ist he absolute temperature in Kelvin, then the number 273.15 has to diminish or T is the temperature in °C. Why should the activation energy be given in kJ/mol, when R is in J/molK?

The Arrhenius approach for Deff is from another theory than Eyring, (which furthermore requires a constant drying rate!) The authors should comment on the differences and why different approaches should be shown here. What ist he aim for the further research?

The authors should evaluate, if the drying rate is really constant, by plotting the drying rate (over time or better over the moisture content (d.b)! In ll. 163-164 the authors tell about a slow down and in ll. 169-170 about the limiting factor for the drying speed. Where do they know about the latter?

  1. 133-135: Misspelling K (for Kelvin) and kB for Boltzmann‘s constant. Why the number of Planck’s constant is not given here, but for kB?

ll.195-215: The authors should comment about the water activity or better the sorption isotherm of kelp. Knowing the sorption isotherm, the equilibrium moisture content (dependent on temperature and air humidity) is clear!

l.241: The drying models should be mentioned and additionally classified (prerequesits, limits…) in the materials and methods section. The link or connection to the Fick-model (equation 4) should be given! In the page model t must be to the power of n not times n! Its main problems are, that the rate constant K for different ns can not be compared, since K has the unit s-n and it can’t be connected to  the Deff-model. So why the authors change again the model when coming from 3.3 to 3.4?

Probably in table 6 the unit oft he enthalpy difference must be kJ/mol (as already stated in the abstract and conclusions.

Due to the manifold problems oft he authors in equations and units, it seems necessary to clarify the presentation and to check and evaluate the mathematical analyses prior to publication.

Author Response

Dear reviewer,

Thank you so much for taking your time to review this manuscript. I appreciate all your comments and suggestions! Please find my itemized responses in below and my revisions in the re-submitted files.

Point 1. It is necessary to describe in more detail the drying arrangement and method: What can be mentioned about the stability (in time) and homogenety (in space) of the air temperature, the air humidity and velocity during the experiments. What is the geometric arrangement of the slices, is the air flowing parallely or is it a cross-flow arrangement….?

Response 1. The drying arrangement and method was supplemented in manuscript (2.2 Heat pump dryer and 2.3 Drying procedure). The air temperature, humidity and velocity were controlled by the controller during the experiments and 4 monitoring points were set around the kelp area in the chamber. The standard deviation of temperature, humidity and air velocity among 4 monitoring points were within ±1.93 ℃, ±2.97%, ±0.04 m/s, respectively. All parameters were stable before drying experiments. The schematic diagram of the heat pump drier and the geometric arrangement of kelp slices was supplemented in Fig.1. The blower provided air from top to bottom through kelp in parallel.

Point 2. The equation (1) seems to be wrong. Either it must be divided by m0 (then it would be on a wet basis) or by the dry mass.

Response 2. The equation (1) was corrected and the subscript was re-checked.

Point 3. In equation (2) and ist explanation it stays unclear, what equilibrium moisture content is used. The equilibrium moisture content is dependent on the air humidity and temperature!

Response 3. The explanation of equation (2) (now equation 3) was supplemented, because the equilibrium moisture content was relatively small and Sappati et al. measured the equilibrium moisture content of kelp at temperature of 40~70 ℃, humidity of 25~50%, and found that the equilibrium moisture content varied from 0.0707 to 0.1270 kg H20/kg dry solids without significantly difference, so in this study, may I use the simplified equation MR=Mt/Mo, four piece of reference papers who used the simplified form were also added here.

Point 4. Equation (4) does not fit to equation (3) since for t=0 equation (4) would not be equal to 1!

Response 4. Equation (4) (now equation 6) was further explained in manuscript (2.6 Effective moisture diffusivity) and transfer to logarithmic form () for easier calculation of Deff. Relevant references were added here.

Point 5. In equation (5) and ist explanation there is a misinterpretation. Either T ist he absolute temperature in Kelvin, then the number 273.15 has to diminish or T is the temperature in °C. Why should the activation energy be given in kJ/mol, when R is in J/molK?

Response 5. The unit mistakes have been corrected.

Point 6. The Arrhenius approach for Deff is from another theory than Eyring, (which furthermore requires a constant drying rate!) The authors should comment on the differences and why different approaches should be shown here. What ist he aim for the further research?

Response 6. The equation (now transferred as )  was used for calculation of Deff, and the equation  was used for calculation of activation energy Ea. The thermodynamic properties were further researched to offer necessary theoretical parameters for drying technology and product quality improvement.

Point 7. The authors should evaluate, if the drying rate is really constant, by plotting the drying rate (over time or better over the moisture content (d.b)! In ll. 163-164 the authors tell about a slow down and in ll. 169-170 about the limiting factor for the drying speed. Where do they know about the latter?

Response 7. The drying rate curves over the moisture content (d.b) was supplemented to support the expression.

Point 8. 133-135: Misspelling K (for Kelvin) and kB for Boltzmann‘s constant. Why the number of Planck’s constant is not given here, but for kB?

Response 8. Misspelling was corrected, and the value of Planck’s constant and kB were supplemented.

Point 9. ll.195-215: The authors should comment about the water activity or better the sorption isotherm of kelp. Knowing the sorption isotherm, the equilibrium moisture content (dependent on temperature and air humidity) is clear!

Response 9. I totally agree that the sorption isotherm of kelp was perfect explanation, but at the beginning of this research, the water activity test was not included, so I’m afraid even though the new kelp will be harvested in months, the experimental data will be different from last year. But considering that the phenomenon was only occurred in 2 groups among 33 groups, which will not exerted significant effects on the conclusions, I prefer to take this valuable suggestion as a highlighting point in next batch of experiment.

Point 10. l.241: The drying models should be mentioned and additionally classified (prerequesits, limits…) in the materials and methods section. The link or connection to the Fick-model (equation 4) should be given! In the page model t must be to the power of n not times n! Its main problems are, that the rate constant K for different ns can not be compared, since K has the unit s-n and it can’t be connected to the Deff-model. So why the authors change again the model when coming from 3.3 to 3.4?

Response 10. The drying models were supplemented with pre-requests, limits and references in the materials and methods section (2.5 Mathematical modeling of drying curves). The mistake in Page model was corrected.

The mathematical modeling aimed for prediction of moisture ratio (3.3 Verification of kinetic model), while the Fick-model (now equation 6) was used for calculation of Deff (3.4 Effective moisture diffusivity). In Table 5 (Effective moisture diffusivity under different drying conditions), the moisture ratio was displayed as lnMR=-5.822×10-5t+ 0.4672, etc, it was not another model of moisture ratio, but a format that easier showing the slope of lnMR for Deff calculation.

Point 11. Probably in table 6 the unit of the enthalpy difference must be kJ/mol (as already stated in the abstract and conclusions.

Response 11. The unit of the enthalpy in Table 6 was corrected as kJ/mol.

Point 12. Due to the manifold problems of the authors in equations and units, it seems necessary to clarify the presentation and to check and evaluate the mathematical analyses prior to publication.

Response 12. The equations and units were re-checked, and the residual analyses were supplemented in mathematical modeling.

Thanks again!

Best regards,

Qian Zhang

Feb 24, 2022

Reviewer 4 Report

The presented manuscript aimed to shorten the drying duration on the premise of uniformity in moisture content by studying the kinetics and thermodynamics of heat pump drying of kelp. The paper has potential; it is interesting and satisfactorily reported; I have the following detailed recommendations for the authors:
Page 1, lines 36-37, The major dehydration method of kelp is nature drying, which is time-consuming and with poor quality -add of drying material, to be more specific,
Page 2, line 56, add reference number after Almeida et al.,
Page 3, where did you place the kelp slices while drying, were they on some trey?
Page 3, For how long, did the drying was performed? Until the 18% moisture of the kelp, but how long? Was the processing time-consuming? Please elaborate. That information can be seen in Figures 1, 2, and 3; however, it should be mentioned in the heat pump drying conditions.
Page 4, lines 129-132, center the formulas for enthalpy change and Gibbs free energy change,
Page 4, consider separating results on results and discussion headings and provide more discussion and comparison with the literature,
Page 9, consider adding residual analysis to the verification of the kinetic model,

Author Response

Dear reviewer,

Thank you so much for taking your time to review this manuscript. I appreciate all your comments and suggestions! Please find my itemized responses in below and my revisions in the re-submitted files.

Point 1. Page 1, lines 36-37, The major dehydration method of kelp is nature drying, which is time-consuming and with poor quality -add of drying material, to be more specific,

Response 1. The drying material, duration and quality effects were supplemented in line 37 with references.

Point 2. Page 2, line 56, add reference number after Almeida et al.,

Response 2. The reference number was added.

Point 3. Page 3, where did you place the kelp slices while drying, were they on some trey?

Response 3. The schematic diagram of the heat pump drier and the geometric arrangement of kelp slices were supplemented in Fig.1. The description was added in 2.3 Drying procedure. During the drying process, two rigid PVC pipes were suspended on the internal bracket, and the kelp slices were hung vertically in 2 rows of 5 slices each with small stainless steel clamps. The blower provided air from top to bottom through kelp in parallel.

Point 4. Page 3, For how long, did the drying was performed? Until the 18% moisture of the kelp, but how long? Was the processing time-consuming? Please elaborate. That information can be seen in Figures 1, 2, and 3; however, it should be mentioned in the heat pump drying conditions.

Response 4. The total drying duration took 4-16h, dependent on the drying conditions and thickness of kelp. This description was added in 2.3 Drying procedure.

Point 5. Page 4, lines 129-132, center the formulas for enthalpy change and Gibbs free energy change,

Response 5. The formulas were centered.

Point 6. Page 4, consider separating results on results and discussion headings and provide more discussion and comparison with the literature,

Response 6. The discussion was supplemented, including the drying rate, activation energy, Gibbs free energy change, etc. But the headings were not separated because I download some published papers from Processes, the format seems like 3. Results and 4 conclusions.

Point 7. Page 9, consider adding residual analysis to the verification of the kinetic model

Response 7. The residual analysis was supplemented to the verification of the kinetic model.

Best regards,

Qian Zhang

Feb 24, 2022

Round 2

Reviewer 3 Report

Although some parts could be further improved, since all points of the reviewer has been addressed in the paper or the letter, it can be published in the present form.